# Genome-Wide Comparative Analysis of Five Amaranthaceae Species Reveals a Large Amount of Repeat Content

**DOI:** 10.3390/plants13060824

**Published:** 2024-03-13

**Authors:** Akshay Singh, Avantika Maurya, Subramani Rajkumar, Amit Kumar Singh, Rakesh Bhardwaj, Surinder Kumar Kaushik, Sandeep Kumar, Kuldeep Singh, Gyanendra Pratap Singh, Rakesh Singh

**Affiliations:** 1Division of Genomic Resources, ICAR-National Bureau of Plant Genetic Resources, Pusa, New Delhi 110012, India; akshaybioinfo@gmail.com (A.S.); avantika.maurya@gmail.com (A.M.); s.rajkumar@icar.gov.in (S.R.); amit.singh5@icar.gov.in (A.K.S.); 2Division of Germplasm Evaluation, ICAR-National Bureau of Plant Genetic Resources, Pusa, New Delhi 110012, India; rakesh.bhardwaj1@icar.gov.in (R.B.); surinder.kaushik1@icar.gov.in (S.K.K.); sandeep.kumar2@icar.gov.in (S.K.); 3International Crop Research Institute for the Semi-Arid Tropics, Hyderabad 502324, India; kuldeep.singh4@icar.gov.in; 4ICAR-National Bureau of Plant Genetic Resources, Pusa, New Delhi 110012, India; gp.singh@icar.gov.in

**Keywords:** comparative analysis, repetitive elements, single-copy orthologs, phylogenetic analysis, LTR elements

## Abstract

Amaranthus is a genus of C4 dicotyledonous herbaceous plant species that are widely distributed in Asia, Africa, Australia, and Europe and are used as grain, vegetables, forages, and ornamental plants. Amaranth species have gained significant attention nowadays as potential sources of nutritious food and industrial products. In this study, we performed a comparative genome analysis of five amaranth species, namely, *Amaranthus hypochondriacus*, *Amaranthus tuberculatus*, *Amaranthus hybridus*, *Amaranthus palmeri*, and *Amaranthus cruentus*. The estimated repeat content ranged from 54.49% to 63.26% and was not correlated with the genome sizes. Out of the predicted repeat classes, the majority of repetitive sequences were Long Terminal Repeat (LTR) elements, which account for about 13.91% to 24.89% of all amaranth genomes. Phylogenetic analysis based on 406 single-copy orthologous genes revealed that *A. hypochondriacus* is most closely linked to *A. hybridus* and distantly related to *A. cruentus*. However, dioecious amaranth species, such as *A. tuberculatus* and *A. palmeri*, which belong to the subgenera Amaranthus Acnida, have formed their distinct clade. The comparative analysis of genomic data of amaranth species will be useful to identify and characterize agronomically important genes and their mechanisms of action. This will facilitate genomics-based, evolutionary studies, and breeding strategies to design faster, more precise, and predictable crop improvement programs.

## 1. Introduction

Amaranths belong to the genus *Amaranthus* L., a historically important, ancient paleopolyploid, C4 dicotyledonous herbaceous plant species that is made up of about 400 species, of which few are found worldwide [1,2]. It belongs to the order Caryophyllales, family Amaranthaceae, subfamily Amaranthoideae, and exhibits disomic inheritance (2*n* = 32) [3]. Approximately 60 Amaranthus species are native to America, while the remaining originated in Asia, Africa, Australia, and Europe. The genus *Amaranthus* contains cultivated wild as well as weedy species. Cultivated amaranths are used for grain, vegetables, forages, and ornamental plants, but food grain and leafy vegetables are the most ancient uses [4]. Among the cultivated species, grain amaranths are one of the domesticated species that have been grown for over 8000 years in the region of Mesoamerica and the Andes mountains [5]. The cultivated grain amaranths include *A. hypochondriacus*, *A. cruentus*, *A. caudatus*, and *A. edulis*, associated with wild species *A. hybridus*, *A. quitensis*, and *A. powellii*. Amaranth species such as *A. hypochondriacus*, *A. cruentus*, and *A. caudatus* are usually domesticated for grain production and are commonly referred to as ‘pseudo-cereals’, and species such as *A. tricolor*, *A. dubius*, *A. blitum*, and *A. viridis* are mostly cultivated as leafy vegetables, while *A. palmeri* (palmer amaranth), *A. retroflexus* (redroot pigweed), *A. spinosus* (spiny amaranth), and *A. albus* (tumbleweed) represent weed species [6,7]. Grain amaranths are known for their magnificent appearance, a few common names of grain amaranths include, *A. hypochondriacus* known as prince’s feather, *A. cruentus* known as purple amaranth, *A. caudatus* known as love-lies-bleeding, which is grown primarily as an ornamental, and *A. tricolor*, known as tampala, grown mostly for its attractive color of leaves [8].

Researchers and consumers are showing a renewed interest in amaranth as a nutri-cereal, particularly among health-conscious populations facing modern-world lifestyle diseases such as diabetes and hypertension [9]. This ancient crop, also known as a ‘millennium crop’, boasts significant nutritional and agronomic versatility [10]. Amaranth grains are rich in protein, minerals (calcium, magnesium, copper, sodium, iron, phosphorus, and zinc), and vitamins (thiamine, riboflavin, ascorbic acid, and niacin) [11,12]. They also contain betacyanin pigment, contributing to the attractive color of amaranth leaves. Additionally, phytochemical analysis of dried amaranth grains reveals alkaloids, phenolics, flavonoids, and saponins, holding potential nutritional and medicinal benefits [13]. The crude protein content varies between 12.5% and 22.5%, surpassing by more than 50% the levels found in the old-world grain crops like wheat (*Triticum aestivum*), rice (*Oryza sativa*), and maize (*Zea mays* L.). Amaranth’s high protein content, gluten-free nature, and balanced amino acid profile make it a valuable dietary option for those suffering from coeliac disease. Moreover, amaranth seed or seed oil is a rich source of vitamin E and squalene, offering potential relief to people with hypertension or cardiovascular disease by lowering blood pressure and cholesterol levels and improving antioxidant status [14,15]. It also reduces the risk of prostate cancer, anemia, osteoporosis and maintains the immune system. The mineral content, particularly potassium, provides additional health benefits, such as preventing hypotension and strengthening respiratory and muscular functions. Flavonoids and phenolic acids present in amaranth grains have biological significance with potential health implications [16]. Amaranth (*Amaranthus* spp.) is indeed a fascinating and versatile plant that has garnered increasing attention due to genetic diversity, adaptability to environments, and resistance to drought, heat, salinity, and pests, making amaranth a promising agricultural crop for addressing food security and agricultural sustainability challenges [17,18]. It provides an alternative or complementary source of nutrition and income in areas with challenging growing conditions and could be a lifesaver for the people suffering from malnutrition and hunger in most developing countries.

Comparative genomics has been transformed by advancements in Next-Generation Sequencing (NGS) technologies, enhancing our understanding of gene and genome structures, dynamics, and functions. This innovative technology has made it possible for researchers to carry out a great deal of research and investigate the intricacies of genetic information in previously unheard-of ways. NGS, with its high-throughput capability and cost-effectiveness, has become an indispensable tool in various fields, ranging from basic biology to clinical diagnostics [19]. The genus Amaranthus comprises numerous species with diverse biological characteristics and potential economic applications. Understanding the genomic variations among these species can shed light on their genetic diversity and evolutionary history. The availability of complete genome sequences of several amaranth species (*A. hypochndriacus*, *A. tuberculatus*, *A. palmeri*, *A. hybridus*, and *A. cruentus*) provides an opportunity to perform comparative genome analysis. Here, we present a comprehensive analysis of key genomic features across five amaranth species, encompassing repetitive element analysis, structural and functional annotation of protein-coding genes, gene family analysis, and phylogenetic distribution based on single-copy orthologs across Amaranthaceae species. Additionally, we investigate the comparative distribution of SSRs and SNPs in both genic as well as intergenic regions of the genomes, transcription factor families, putative miRNAs, and transporter genes.

### Characteristics of Amaranth Species

Grain amaranth (*A. hypochondriacus*) is a versatile plant with applications in food, feed, vegetables, and ornamental purposes. Commonly known as Prince’s feather, Mexican grain amaranth, pigweed, or ramdanna, this annual herb features an erect stem and broad, lance-shaped leaves. The plant is cultivated extensively in India, particularly in the sub-Himalayan ranges and the Nilgiri Hills of South India. The small seeds serve both culinary and ornamental purposes, and the plant is prized for its edible leaves [20]. *A. tuberculatus*, known as rough-fruited amaranth, rough-fruited water-hemp, tall water-hemp, or rough pigweed, is an annual weed characterized by a robust taproot and a bushy growth habit. Thriving in agricultural fields and disturbed areas demonstrates resistance to specific herbicides [21]. *A. palmeri*, commonly referred to as palmer’s amaranth or careless weed, stands out as a fast-growing, competitive annual weed. Recognized as one of the most invasive species among dioecious amaranths, it poses significant challenges to various U.S. crops due to its rapid development of herbicide resistance [21]. *A. hybridus*, or smooth pigweed, is a smooth-stemmed annual plant native to tropical and subtropical America. Its lance-shaped leaves and adaptability to diverse habitats, including cultivated fields, gardens, orchards, and disturbed areas, contribute to its status as a problematic weed in agriculture [21]. *A. cruentus*, an annual herbaceous flowering plant originating from Central America, is cultivated for its seeds used as a pseudo-cereal. Also known as purple amaranth, it has been grown since ancient times for both its ornamental and culinary values. The edible leaves and versatile applications make it a noteworthy plant in various agricultural and horticultural contexts [22] (Figure 1).

## 2. Results

### 2.1. Genomic Structure and Evaluation of Selected Amaranth Genomes

Each of the complete genome structures of *A. hypochondriacus*, *A. tuberculatus*, *A. palmeri*, *A. hybridus*, and *A. cruentus* has been represented in the form of scaffolds and chromosome-wise circular structures using the Circos-0.69 software [23]. The genomic features such as the distribution of SNPs present in the genic region (including exon, intron, 5′UTRs, and 3′UTRs); SNPs in the intergenic region; density distribution of TFs; distribution of protein-coding genes; dispersal of mono- to hexa-type SSRs; transporter genes; putative miRNAs are shown in Figure 2. To assess the completeness of each of the five amaranth genomes, we used the Benchmarking (Universal Single-Copy Orthologs BUSCO) plant lineage dataset. From the 2326 core eudicot genes, 2198 (94.5%), 2228 (95.79%), 2208 (94.93%), 2233 (96%), and 2154 (92.61%) were identified in the *A. hypochondriacus*, *A. tuberculatus*, *A. palmeri*, *A. hybridus*, and *A. cruentus* assemblies, respectively, with 2100 (90.3%), 2068 (88.9%), 1921 (82.6%), 2131 (91.6%), and 2063 (88.7%) complete single-copy genes, respectively (Table 1). Among the surveyed genomes, *A. hybridus* had the highest BUSCO score with 2233 complete BUSCOs (96%); another 0.6% of sequences were fragmented (14 BUSCOs); 3.4% were considered missing (79 BUSCOs). For evaluation and completeness of genes in the assemblies, unigenes generated from the transcriptomic data of different amaranth species were mapped with the amaranth genome assemblies. The results indicated that each genome assembly covered about 84–90% of the expressed unigenes, suggesting that the assembled genomes contained a high percentage of expressed genes.

### 2.2. Repetitive Elements Analysis

Repeat Modeler and Repeat Masker were used to identify and annotate the repetitive components in five amaranth species. The repeat analysis pipeline total numbers were 229,717,046 (56.88% of the total genome size), 435,846,425 (63.26%), 224,470,421 (54.49%), 233,584,532 (56.72%), and 207,438,445 (56.80%) non-redundant repetitive sequences in *A. hypochondriacus*, *A. tuberculatus*, *A. palmeri*, *A. hybridus*, and *A. cruentus*, respectively. The percentage of repetitive sequences predicted and genome sizes do not correlate to each other, as the genome sizes of *A. hypochondriacus* and *A. cruentus* are lesser than those of *A. palmeri* and *A. hybridus*, while repetitive sequences are greater in *A. hypochondriacus* and *A. cruentus* as compared to *A. palmeri* and *A. hybridus*. The major classes of repeat elements predicted in the amaranth genomes are shown in Table 2. Table 2 shows the composition of the repetitive portion of each genome in terms of repeat classes: Short-Interspersed-Elements (SINEs), Long-Interspersed-Elements (LINEs), Long Terminal Repeat (LTR) elements, DNA transposons, small RNA, satellite DNAs, simple repeats, and low complexity repeats. Out of the predicted repeat classes, the majority of repetitive sequences were LTR elements, which account for about 19.43%, 24.89%, 17.95%, 13.91%, and 20.94% in *A. hypochondriacus*, *A. tuberculatus*, *A. palmeri*, *A. hybridus*, and *A. cruentus*, respectively. LTRs can be further sub-classified into *Copia*-like and *Gypsy*-like elements, both of which are commonly present in angiosperms. The distribution of *Copia*-like and *Gypsy*-like LTRs exhibited a similar pattern in *A. hypochondriacus* (10.81%, 6.62%), *A. palmeri* (9.62%, 6.23%), *A. hybridus* (7.62%, 5.69%), and *A. cruentus* (11.63%, 7.02%), with *Copia*-like LTRs percentage being significantly higher than *Gypsy*-like, while in *A. tuberculatus*, *Copia*-like LTRs (9.25%) are lower than *Gypsy*-like LTRs (15.03%). The second major category was DNA transposons, which took up 8.42%, 8.0%, 6.66%, 9.1%, and 8.45% of each genome, respectively (Table 2).

### 2.3. Gene Content, Distribution, and Functional Annotation

A total of 170,477 protein-coding genes were predicted in five amaranth species. The highest number of genes was present in the *A. palmeri* genome (48,625), followed by *A. cruentus* (43,382), *A. tuberculatus* (30,771), *A. hypochondriacus* (23,883), and *A. hybridus* genome (23,820), respectively. The genes were distributed throughout the sixteen scaffolds in *A. hypochondriacus*, *A. tuberculatus*, *A. hybridus*, and *A. palmeri*, while in *A. cruentus*, genes were distributed in seventeen chromosomes, as chromosome 2 was disseminated into chromosomes 2A and 2B. Scaffold 1 contains the highest number of genes, approximately 9.2–10.4% of the total genes, and scaffold 16 possesses the least number of genes, which covers approximately 3.4–4.2% of the total genes (Figure 2). Compared to other sequenced species of the same family, *C. quinoa* [24] has more genes than *A. hypochondriacus*, *A. tuberculatus*, *A. hybridus*, and *A. cruentus* but fewer than *A. palmeri*. While *S. aralocaspica* [25], *S. oleracea* [26], and *B. vulgaris* [27] contain genes in a similar range as *A. hypochondriacus*, *A. tuberculatus*, *A. hybridus*, and *A. cruentus*. The mean gene as well as CDS length vary from 4154 to 4854 and 941 to 1486 bp, respectively (Table 3). Numerous gene structure features, such as mean exons per gene, mean exon length (bp), mean introns per gene, and mean intron length (bp), were compared. On average, protein-coding genes in *A. hypochondriacus* are 4102 bp long, contain 5 exons, and have a mean exon length of 219 bp, with these values similar to those of other sequenced Amaranthaceae species except for *S. oleracea*, which has a much higher mean gene length of 5716 bp. The ranges of mean CDS length, mean exon length, and mean intron length are 941–1486 bp, 219–385 bp, and 394–937 bp, respectively (Table 3).

Out of the total protein-coding genes, 95.28% in *A. hypochondriacus*, 88.75% in *A. tuberculatus*, 88.20% in *A. hybridus*, 88.05% in *A. palmeri*, and 89.10% in *A. cruentus* were functionally annotated (Table 4). Of the unannotated genes, 1126, 3462, 5736, 2847, and 4728 were specific to *A. hypochondriacus*, *A. tuberculatus*, *A. palmeri*, *A. hybridus*, and *A. cruentus*, respectively. In *A. hypochondriacus* and *A. palmeri*, the majority of genes are annotated with the NCBI-NR database, followed by TrEMBL, InterProScan, Swiss-Prot, COG, KEGG, and GO databases. While in *A. tuberculatus*, *A. hybridus*, and *A. cruentus*, the maximum number of genes is annotated with the NCBI-NR database, followed by TrEMBL, InterProScan, Swiss-Prot, COG, GO, and KEGG databases, respectively. The functional annotation of the genes allows us to classify genes into different functional classes, which can be very useful in determining the physiological significance of a large number of genes. Gene annotation provides insight into the molecular aspects of amaranth genes based on comparison with other species.

### 2.4. Gene Family Construction and Phylogenetic Distribution

Protein sequences from five amaranth species and five other species (*A. thaliana*, *B. vulgaris*, *C. quinoa*, *S. oleracea*, and *S. aralocaspica*) were retrieved to construct gene families based on an all-vs.-all alignment with an E-value cutoff of 1 × 10^−5^. A total of 43, 502, 1004, 174, and 567 gene families were identified specific to *A. hypochondriacus*, *A. tuberculatus*, *A. palmeri*, *A. hybridus*, and *A. cruentus*, which contain 102, 1609, 3116, 411, and 1851 genes, respectively (Appendix A). Furthermore, the 9655 gene families of *A. hypochondriacus*, *A. tuberculatus*, *A. palmeri*, *A. hybridus*, and *A. cruentus* were clustered (Figure 3A), of which 203 orthologous gene families containing 269 genes were specific to *A. hypochondriacus*, 763 gene families containing 1928 genes were specific to *A. tuberculatus*, 1135 gene families containing 3268 genes were specific to *A. palmeri*, 341 gene families containing 637 genes were specific to *A. hybridus*, and 706 gene families containing 2021 genes were specific to *A. cruentus*. Moreover, 9525 gene families of *A. thaliana*, *B. vulgaris*, *C. quinoa*, *S. aralocaspica*, and *S. oleracea* were clustered (Appendix A), of which 1378 orthologous gene families containing 5560 genes were specific to *A. thaliana*, 726 gene families containing 1636 genes were specific to *B. vulgaris*, and 1420 gene families containing 5148 genes were specific to *C. quinoa*, respectively. Of all the orthologous gene families predicted for a total of 10 species, 406 gene families are composed of single-copy orthologs containing one representative protein sequence of each species. These sequences were used to reconstruct a phylogenetic tree (Figure 3B), which depicts the phylogenetic relationship among amaranth as well as other species. This analysis shows that *A. hypochondriacus* is more closely related to *A. hybridus* and *A. cruentus* than *A. palmeri* and *A. tuberculatus*, as expected. And *C. quinoa* appeared to be more closely related to *S. oleracea* (Spinach) than *B. vulgaris*, as expected from earlier molecular analyses [28].

KEGG pathway enrichment analysis for orthologous genes of five amaranth species was also conducted (Appendix A). Functional annotation revealed that in *A. hypochondriacus*, these orthologs corresponded mainly with carbohydrate metabolism, lipid metabolism, energy metabolism, metabolism of other amino acids, and biosynthesis of other secondary metabolites. However, for *A. tuberculatus*, the metabolism of cofactors and vitamins, as well as the metabolism of terpenoids and polyketides were enriched, while in *A. cruentus*, the pathways corresponding to glycan biosynthesis and metabolism, the metabolism of cofactors and vitamins, and the metabolism of other amino acids were enriched. In *A. palmeri*, enrichment occurred in carbohydrate metabolism, lipid metabolism, metabolism of other amino acids, and biosynthesis of other secondary metabolite pathways. In *A. hybridus*, pathways related to lipid metabolism, carbohydrate metabolism, energy metabolism, and amino acid metabolism were enriched. Using Gene Ontology (GO) analysis, ortholog genes in *A. hypochondriacus*, *A. tuberculatus*, *A. palmeri*, *A. hybridus*, and *A. cruentus* were enriched (Appendix A). The top five GO annotation categories of *A. hypochondriacus* genes were C: membrane, C: nucleus, P: regulation of DNA-templated transcription, F: metal ion binding, F: ATP binding; for *A. tuberculatus* genes, F: nucleic acid binding, C: nucleus, C: membrane, F: RNA binding, F: metal ion binding; for *A. palmeri*, F: zinc ion binding, F: nucleic acid binding, C: membrane, P: DNA integration, F: metal ion binding; for *A. hybridus*, C: membrane, F: nucleic acid binding, F: metal ion binding, C: nucleus, F: ATP binding; for *A. cruentus*, F: zinc ion binding, F: nucleic acid binding, F: metal ion binding, C: membrane, P: DNA integration, respectively.

### 2.5. Comprehensive Distribution of Microsatellites

The comprehensive identification and comparative analysis of perfect SSRs in five amaranth genomes was carried out in this study. A total of 838,579 perfect SSRs containing mono- to hexa-nucleotide types of repeats were identified from 2281.82 Mb of genome sequences in five amaranth species, with an average relative abundance (loci/Mb) and relative density (bp/Mb) of 367.5 and 12,815.54, respectively (Table 5). The results demonstrated that the highest number of SSR markers was identified in *A. hypochondriacus* (243,288), followed by *A. tuberculatus* (216,733), *A. palmeri* (144,801), *A. hybridus* (132,717), and *A. cruentus* (101,040). The higher microsatellite’s relative abundance (bp/Mb) and relative density (SSRs/Mb) were 602.43 and 16,894.21, respectively, as observed in *A. hypochondriacus*, whereas the lower was observed in *A. cruentus*, which is 276.77 and 8707.26 (Table 5).

From the total SSRs predicted among five amaranth species, di-nucleotides were the most abundant, followed by tri-nucleotides, mono-nucleotides, tetra-nucleotides, penta-nucleotides, and hexa-nucleotide repeats, except in *A. hybridus*, *A. palmeri*, and *A. cruentus*, which had greater numbers of hexa-nucleotides compared to penta-nucleotide repeats (Figure 4; Appendix A).

Among the different types of repeats present in all five amaranth species, it was observed that in each motif type, one particular motif was prevalent. From all the identified SSRs, (14.68–27.08%) of the total mono-nucleotide repeats were ‘A’, (39.14–61.22%) of the total di-nucleotide repeats were ‘AT’, (12.15–18.53%) of the total tri-nucleotide repeats was ‘AAT’, (0.74–1.4%) of the total tetra-nucleotide repeats was ‘AAAT’, (0.11–0.44%) of the total penta-nucleotide repeats was (‘AAAAT’, ‘ATCAG’), and (0.05–0.17%) of the total hexa-nucleotide repeats was (‘AATAAC’, ‘AAAAAT’, ‘ATATAC’) (Appendix A). Out of the total SSRs predicted for all five amaranth species, a higher number of three sets of primer pairs were generated for *A. tuberculatus* (77,814), followed by *A. hypochondriacus* (57,404), *A. hybridus* (50,059), *A. palmeri* (41,930), and *A. cruentus* (36,462), respectively (Table 6). Distribution-wise, mono-nucleotide repeat primer pairs were the most abundant, followed by di, tri, tetra, hexa, and penta-nucleotides in *A. hybridus*. In the case of *A. tuberculatus*, mono-nucleotide repeat primer pairs were the most abundant, followed by tri, di, tetra, penta, and hexa-nucleotide. Whereas in *A. palmeri* and *A. cruentus*, mono-nucleotide repeat primer pairs were the most abundant, followed by tri, di, tetra, hexa, and penta-nucleotides (Table 6), and in *A. hypochondriacus*, di-nucleotide repeat primer pairs were the most abundant, followed by mono, tri, tetra, hexa, and penta-nucleotides. These SSR primers can be used for candidate gene identification, linkage mapping, genetic diversity analysis, and phylogenetic relationships among amaranth species.

The SSR motifs were not uniformly distributed in the genic as well as intergenic regions. The majority of SSRs were found in the intergenic region of the genome. It accounts for *A. hypochondriacus* (43,156; 75.17%), *A. tuberculatus* (62,898; 80.83%), *A. hybridus* (40,137; 80.17%), *A. palmeri* (23,955; 57.13%), and *A. cruentus* (26,951; 73.91%) of the total identified SSRs (Appendix A). The intergenic regions had the most abundant SSRs in the pattern: Intergenic > Introns > Exons > 5′UTRs > 3′UTRs in three amaranth species except for *A. hybridus* and *A. tuberculatus*, which followed the pattern Intergenic > Introns > Exons > 3′UTRs > 5′UTRs (Appendix A). In the exonic regions, tri-nucleotide SSRs were the most frequent type, followed by di > hexa > mono > tetra > penta-nucleotide SSRs for all five amaranth species (Figure 5A). Whereas, in the intronic region, mono-nucleotide SSRs were the most frequent type, followed by di > tri > tetra > penta > hexa-nucleotide SSRs for all five amaranth species (Figure 5B). In the 5′UTRs, 3′UTRs, and intergenic regions, mono-nucleotide SSRs were the most frequent type, followed by the pattern: di > tri > tetra > hexa > penta-nucleotide SSRs in all five amaranth species except for *A. hypochondriacus*, which followed the pattern di-nucleotide > mono > tri > tetra > hexa > penta-nucleotide (Figure 5C–E).

### 2.6. Sequence Variants (SNPs)

DNA-based markers such as SNP markers have gained popularity due to their abundance and their role in marker-assisted selection. The BioProject numbers PRJNA290898, PRJNA432348, and PRJNA626536 have been used for SNP mining. A total of 958,646, 1,570,771, 760,209, 997,675, and 1,322,409 SNPs were identified in *A. hypochondriacus*, *A. tuberculatus*, *A. palmeri*, *A. hybridus*, and *A. cruentus*, respectively. SNPs were distributed asymmetrically across all sixteen scaffolds. The highest number of SNPs in *A. hypochondriacus* is found in scaffold 6, 106,528 (11.11%), followed by scaffold 8 (9.71%), scaffold 1 (9.70%), scaffold 9 (8.37%), scaffold 5 (8.26%), scaffold 3 (8.17%), and scaffold 12 (1.54%) in that order (Figure 6). On the other hand, scaffold 13 in *A. tuberculatus* contains the highest number of SNPs, 160,174 (10.19%), followed by scaffold 1 (10%), scaffold 6 (9.88%), scaffold 2 (8.01%), scaffold 3 (7.06%), scaffold 4 (6.93%), and scaffold 16 (3.52%) in that order (Figure 6). In *A. palmeri*, scaffold 4 has the highest number of SNPs, 91,732 (12.06%), followed by scaffold 1 (10.54%), scaffold 2 (8.97%), scaffold 5 (7.73%), scaffold 6 (6.58%), scaffold 3 (6.57%), and scaffold 7, which contains the least number of SNPs, 16,358 (approximately 2.15%) of the total SNPs (Figure 6). While in *A. hybridus*, scaffold 1 contains the highest number of SNPs, 106,720 (10.76%), followed by scaffold 2 (9.31%), scaffold 4 (8.2%), scaffold 3 (7.14%), and scaffold 5 (6.71%). Similarly, in *A. cruentus*, scaffold 1 contains the highest number of SNPs, 142,684 (10.79%), followed by scaffold 2 (9.51%), scaffold 4 (7.9%), scaffold 9 (7.5%), and scaffold 3 (7.0%) (Figure 6).

Among the total 958,646 (*A. hypochondriacus*), 1,570,771 (*A. tuberculatus*), 760,209 (*A. palmeri*), 997,675 (*A. hybridus*), and 1,322,409 (*A. cruentus*) SNPs, 21.97%, 22.50%, 16.52%, 23.07%, and 21.89% were found to be in genic regions, and 78.03%, 76.85%, 83.47%, 76.93%, and 78.11% were in intergenic regions, respectively (Appendix A). The SNPs present in the genic region were further classified based on the gene architecture, i.e., exon, intron, 5′UTRs, and 3′UTRs. Out of the total genic SNPs, the maximum number of SNPs is present in the intron region, followed by exons > 3′UTRs > 5′UTRs (Appendix A). Allelic variation within economically important genes can be identified using genic SNPs and can be used successfully in planning crop improvement programs.

### 2.7. Transcription Factors (TFs)

Transcription Factors (TFs) play an important role in gene expression and can be utilized in functional and evolutionary studies. Both wild and cultivated amaranth species have a marked ability to tolerate diverse abiotic and biotic stresses such as drought, salinity, higher temperatures, defoliation, and pathogens; thus, the study of transcription factors can provide insight into stress-related genes. Transcriptomic analysis of the amaranth species has proven to be an effective method for identifying genes that are activated in response to these stress conditions. TFs play an important role in plant responses to biotic and abiotic stresses by altering the expression of several responsive genes. In recent years, several amaranth genes of unknown function and transcription factors have been functionally characterized [28,29]. For each species, the total predicted TFs were categorized into 57 families based on their domain sequence. The highest number of TFs was predicted in *A. palmeri* (19,481), followed by *A. cruentus* (17,665), *A. tuberculatus* (8685), *A. hypochondriacus* (7163), and *A. hybridus* (6655). Some stress-related important transcription factor families, namely, bHLH, NAC, AP2/ERF, WRKY, bZIP, C2H2, Dof, and MYB, have been studied in amaranth. In *A. hypochondriacus*, out of the total significant matches, the maximum number of protein-coding genes was categorized into bHLH (10.54%), followed by ERF (7.43%), NAC (7.37%), MYB-related (6.04%), WRKY (5.82%), M-type (4.17%), MYB (3.77%), C2H2 (3.77%), bZIP (3.66%), and FAR1 (3.31%) transcription factor families (Figure 7A; Appendix A). Similarly, in *A. tuberculatus*, the maximum number of TFs belongs to bHLH (9.65%), followed by ERF (7.7%), NAC (6.48%), MYB-related (6.17%), WRKY (5.09%), M-type (4.81%), TCP (4.76%), MYB (4.01%), FAR1 (3.8%), and bZIP (3.55%) transcription factor families (Figure 7B; Appendix A). In *A. hybridus*, most of the TFs belong to bHLH (10.59%), followed by NAC (7.93%), ERF (7.17%), MYB-related (6.0%), WRKY (5.42%), M-type (4.13%), FAR1 (3.88%), bZIP (3.86%), C2H2 (3.65%), and MYB (3.56%) transcription factor categories (Figure 7C; Appendix A). In the species *A. palmeri*, the maximum number of transcription factor belongs to LBD (10.52%), followed by ERF (9.79%), M-type (8.94%), TCP (7.54%), NAC (6.33%), MYB-related (5.77%), bHLH (5.67%), MYB (4.42%), WRKY (4.01%), and E2F-DP (3.89%) transcription factor families (Figure 7D; Appendix A). While in *A. cruentus*, the majority of TFs belong to M-type (10.89%), followed by ERF (8.73%), TCP (7.82%), LBD (7.55%), bHLH (6.63%), MYB-related (6.2%), NAC (5.75%), WRKY (4.49%), MYB (4.44%), and FAR1 (4.24%) transcription factor families (Figure 7E; Appendix A).

The distribution pattern of TFs across scaffolds showed that scaffold 1 had the highest numbers of TFs, followed by scaffold 2, scaffold 4, scaffold 3, and the lowest numbers of TFs present in scaffold 16 in *A. hypochondriacus*, *A. tuberculatus*, and *A. hybridus* species (Appendix A). In *A. palmeri*, the highest numbers of TFs were present in scaffold 2, followed by scaffold 4, scaffold 1, and scaffold 5, and the lowest numbers of TFs were present in scaffold 7 (Appendix A). In *A. cruentus*, the maximum number of TFs was present in scaffold 2, followed by scaffold 1, scaffold 3, scaffold 4, and the minimum number of TFs was present in scaffold 16 (Appendix A).

### 2.8. microRNAs and Transporter Genes

Recent genome sequencing of the amaranth species *A. hypochondriacus*, *A. tuberculatus*, *A. hybridus*, *A. palmeri*, and *A. cruentus* has enabled the discovery of miRNAs, which can be used as biotechnological tools in amaranth breeding. MicroRNA-guided gene silencing has become a significant mode of gene regulation in plants at the transcriptional and post-transcriptional levels. A total of 123 pre-miRNAs encoding 31 mature miRNAs belong to 21 miRNA families in *A. hypochondriacus*, 94 pre-miRNAs encoding 27 mature miRNAs belong to 22 miRNA families in *A. tuberculatus*, 113 pre-miRNAs encoding 32 mature miRNAs belong to 24 miRNA families in *A. hybridus*, 109 pre-miRNAs encoding 31 mature miRNAs belong to 24 miRNA families in *A. palmeri*, and 80 pre-miRNAs encoding 32 mature miRNAs belonging to 25 miRNA families in *A. cruentus* were predicted (Table 7).

The length of the pre-miRNAs ranged from 90 to 251 nucleotides with an average of 135 nucleotides in the *A. hypochondriacus* genome, from 88 to 387 nucleotides with an average of 136 nucleotides in the *A. tuberculatus* genome, from 86 to 314 nucleotides with an average of 138 nucleotides in the *A. hybridus* genome, from 88 to 248 nucleotides with an average of 135 nucleotides in the *A. palmeri* genome, and from 88 to 239 nucleotides with an average of 139 nucleotides in the *A. cruentus* genome, respectively (Appendix A). The mature miRNAs were 20–24 nucleotides in length, with a modal number of 21 nucleotides in five amaranth genomes (Appendix A). Out of the total predicted pre-miRNAs, the maximum number of miRNAs belongs to the miR466 family, followed by miR157, miR169, miR399, miR166, miR167, and miR171, respectively, in five amaranth species (Appendix A). It is necessary to identify the miRNA-targeted genes to know the functions of predicted miRNAs. Using psRNATarget, we predicted a total of 512, 869, 707, 659, and 365 target genes in the *A. hypochondriacus*, *A. tuberculatus*, *A. hybridus*, *A. palmeri*, and *A. cruentus* genomes, respectively (Appendix A). The target genes of the five amaranth species were functionally annotated and classified into three main categories: (i) biological process, (ii) molecular function, and (iii) cellular component (Appendix A). In the classification of biological processes, out of the total number of genes, the maximum number of genes belongs to the following subcategories: DNA integration, followed by transcriptional regulation, phosphorylation, protein phosphorylation, and proteolysis; under the molecular function category, the maximum number of genes falls under the subcategories: ATP binding, followed by transferase activity, nucleic acid binding, nucleotide binding, and metal ion binding; in the list of cellular components, the maximum number of genes falls under the subtypes: membrane, followed by integral components of membrane, nucleus, cytoplasm, and extracellular regions in the five species of amaranth (Appendix A).

In the transporter gene analysis, out of all the predicted transmembrane proteins in *A. palmeri* (6984), followed by *A. tuberculatus* (6443), *A. cruentus* (5789), *A. hybridus* (5106), and *A. hypochondriacus* (5789), non-redundant transporter genes carrying two or more transmembrane domains, which are (2760) *A. hypochondriacus*, (2905) *A. tuberculatus*, (2642) *A. hybridus*, (3402) *A. palmeri*, and (2760) *A. cruentus*, were selected for further analysis. In the five amaranth species, the maximum number of transporter genes contained two TM domains, gradually decreasing as the number of TM domains increased (Figure 8).

The classification of the identified transporter genes was performed according to the Transporter Classification Database (TCDB) system into well-defined classes such as (i) channels/pores, (ii) electrochemical potential-driven transporters, (iii) primary active transporters, (iv) group translocators, (v) transmembrane electron carriers, (viii) accessory factors involved in transport, and (ix) incompletely characterized transport systems (Appendix A). Among the seven types of transporter classes, the major transporter genes belong to the category electrochemical potential-driven transporter (34.13–36.21%), followed by channels/pores (19.72–23.13%), incompletely characterized transport systems (16.8–18.36%), primary active transporters (14.46–16.36%), accessory factors involved in transport (5.88–8.19%), group translocators (2–2.22%), and transmembrane electron carriers (1.15–1.49%) in the five amaranth species (Appendix A).

The super-family enrichment analysis of the predicted transporter genes showed that the Major Facilitator Super-family (MFS; TCDB: 2.A.1) is the most abundant super-family, followed by the Amino Acid-Polyamine-Organocation Family (APC; TCDB: 2.A.3), ATP-binding Cassette Super-family (ABC; TCDB: 3.A.1), Drug/Metabolite Transporter Super-family (DMT; TCDB: 2.A.7), and Mechano-sensitive Calcium Channel Family (MCA; TCDB: 1.A.87) (Appendix A). The major facilitator superfamily (MFS) is one of the largest classes of transporter proteins and is essential for the movement of various substrates across biological membranes. In the transport mechanism, the MFS transporter protein undergoes a series of conformational changes that allow for the passage of molecules from the extracellular fluid to the cytoplasm and vice versa [30].

## 3. Discussion

In this study, five complete genome sequences of the genus Amaranth, including *A. hypochondriacus* [3], *A. tuberculatus*, *A. palmeri*, *A. hybridus* [21], and *A. cruentus* [22], were taken for comparative genomic study. The genome sizes ranged from 365.20 Mb (*A. cruentus*) to 688.98 Mb (*A. tuberculatus*), similar to the genome size of *S. aralocaspica* [25]. *A. tuberculatus* has a larger genome size as compared to other species, indicating polyploidization during an ancestral split. The variation in genome sizes suggests a random evolution with independent genome duplication and chromosome loss, fusion, and fission events. The GC% of the genomes were very similar, ranging from 32.3 to 37.18% (Table 8), which has also been observed in other related genomes such as *C. quinoa* [24], *B. vulgaris* [27], and *S. aralocaspica* [25]. Genome features, including genome size, repeat content, heterozygosity, polyploidy, and GC percentage, have been shown to affect the quality of de novo assemblies; hence, genome profiling offers important insight into achieving a high-quality genome assembly [31]. The assembly completeness analysis showed that amaranth genomes present 92.61–96% of all BUSCOs (analysis with eudicot database) (Table 2), and, compared with the other sequenced genomes currently available (*S. aralocaspica*; 89.5%) [25], (*S. oleracea*; 97.2%) [32], (*C. quinoa*; 97.3%) [24], it represents the good quality assemblies of amaranth genomes. Overall, single-copy ortholog (BUSCO) analysis revealed that all five amaranth genome assemblies were quite complete and contained most of the sequenced genes in each species. This means that even if repeat regions collapsed to make genomes compact, the majority of core orthologs were captured in the assemblies.

The analysis of repetitive elements in the genomes of these amaranth species sheds light on their distribution and overall composition. Repetitive DNA elements are a major part of the nuclear genome, and thus are essential for understanding the regulation and evolution of the genome. Repeat analysis results of *A. hypochondriacus* (56.88%), *A. tuberculatus* (63.26%), *A. palmeri* (54.49%), *A. hybridus* (56.72%), and *A. cruentus* (56.80%) are in a similar range to the previously sequenced genomes of *C. quinoa* [24] and S. *oleracea* [32]. The predominance of Long Terminal Repeat (LTR) elements and DNA transposons suggests that these elements have played a significant role in genome evolution; these predictions are comparable to earlier studies [24,25,27]. LTR elements are often associated with the regulation of gene expression and could potentially contribute to genome size variations across these species. The most striking observation from this analysis is the lack of a straightforward correlation between the percentage of repetitive sequences and the total genome size in these amaranth species. *A. hypochondriacus* and *A. cruentus*, despite having smaller genome sizes, exhibit a higher percentage of repetitive sequences compared to *A. palmeri* and *A. hybridus*, which have larger genomes. This discrepancy suggests that the accumulation of repetitive elements in these genomes does not solely depend on genome size. The percentage of repeats in published plant genomes varies greatly, for example, the minute 82 Mb genome of *Utricularia gibba* L. [33] contains only 3% of repeats, while *Zea mays* [34] contains 85% of repeats. The gene distribution and annotation results for the studied amaranth species offer significant insights into the genomic landscape and functional characteristics of these plant species. The variation in the number of protein-coding genes among the five amaranth species (*A. palmeri* (48,625), *A. cruentus* (43,382), *A. tuberculatus* (30,771), *A. hypochondriacus* (23,883), and *A. hybridus* (23,820)) reflects the genetic diversity within the Amaranthaceae family. This estimate is similar to other reported plant species such as *B. vulgaris* (27,421) [27], *S. oleracea* (28,964) [32], and *S. aralocaspica* (29,064) [25]. The highest number of genes in *A. palmeri* suggests a potentially more complex genomic architecture or a higher degree of gene duplication in this species compared to others. The distribution of genes across scaffolds and chromosomes provides valuable information on the genomic organization, with scaffold 1 containing the highest number of genes, indicating a potential hotspot for gene activity. The consistency in gene structure features, such as the number of exons and introns, suggests a degree of evolutionary conservation in the genetic architecture across these species, with *A. hypochondriacus* exhibiting similar characteristics to other sequenced Amaranthaceae species. The higher gene counts *in C. quinoa* and *A. palmeri* compared to other amaranth species may indicate specific adaptations or complexities in their genetic makeup, possibly contributing to unique physiological traits or environmental adaptations. The high percentage of functionally annotated genes (ranging from 88.05% to 95.28%), which is consistent with earlier studies such as *S. aralocaspica* (97.2%) [25] and *S. glauca* (91.80%) [35], is indicative of a comprehensive understanding of the biological roles of these genes. The use of various databases for annotation, such as NCBI-NR, TrEMBL, InterProScan, Swiss-Prot, COG, KEGG, and GO, enhances the robustness of the functional annotations, providing a multi-faceted view of the gene functions. Understanding the genetic makeup and functional roles of these genes can have implications for crop improvement programs, allowing for the development of varieties with enhanced traits such as yield, resistance to diseases, or tolerance to environmental stresses.

The gene clustering and phylogenetic analysis conducted on protein sequences from five amaranth species and five other related plant species provides valuable insights into the evolutionary relationships and genomic divergence among these species. The phylogenetic analysis based on single-copy orthologs provides an overall measure of relatedness between a pair of species with respect to any single gene comparison [36]. Here, we studied the phylogenetic relationship using 406 single-copy orthologs from nine Amaranthaceae species and *Arabidopsis thaliana* as an outgroup. Species clustered into three clades: the first clade comprises two dioecious amaranths, i.e., *A. tuberculatus* and *A. palmeri*, corresponding to the subgenera *Amaranthus acnida* [37]. Both the weedy amaranths bipartitioned within the clade, reflecting phylogenetic divergence due to the polyploid genome of *A. tuberculatus* [38]. Caryophylleae, Betoideae, and Chenopodioideae formed a sister clade within the first clade similar to the previously reported phylogenetic relationships based on 62 protein-coding genes of chloroplasts from 31 taxa by Su-Young Hong, 2018. The second and third clades correspond to the subgenera *Amaranthus*, or hybridus clade. *A. hypochondriacus* and *A. hybridus* were more closely related, as suggested by [39], forming a distinct clade with *A. cruentus*. Similar to our results [40], we delineated that *A. hypochondriacus* and *A. hybridus* could be grouped as leafy and grain amaranth using amplified fragment length polymorphism, and double-primer fluorescent inter simple sequence repeat markers. The results of this analysis contribute to our understanding of the genetic diversity within the Amaranthaceae family and shed light on the evolutionary history of individual species. The varying numbers of species-specific gene families suggest that each species has undergone distinct evolutionary pressures, leading to the acquisition or loss of specific genetic elements. The presence of single-copy orthologs in some gene families indicates evolutionary constraints on these genes, emphasizing their importance in maintaining essential functions across diverse plant lineages. Furthermore, the identified gene families and their clustering patterns serve as a foundation for future functional genomics studies, helping researchers explore the roles of specific genes in the adaptation and development of these plant species.

SSR markers are one of the most important molecular markers and play a significant role in genetic dissection and marker-assisted crop breeding due to their abundance, reproducibility, and polymorphism. SSRs are classified as simple and compound types. The perfect SSRs with a length of 20 bases or above contain continuous repetitions without any interruption, and compound SSRs are interrupted by a non-repetitive nucleotide sequence with a length of 100 bases [41]. In addition, SSRs were identified within Amaranthaceae, as it proved to be a stepping stone for the genetic dissection of complex traits for crop enhancement and varietal development. Genome-wide SSRs were mined as they provide insight into gene regulation and genome organization; these are highly polymorphic, facilitating better map coverage [42,43]. It was observed that intergenic SSRs were more abundant than genic SSRs. Within the genic region, SSRs had a higher quantity and density in the CDS, or exonic region, indicating strong selection pressure. The A/T motif was prominent in di-nucleotide repeats, indicating the presence of poly (A) tails of densely scattered retroposed sequences [44]. *A. hypochondriacus* consists more of di-nucleotides, whereas mono-nucleotide repeats are abundant in other species. In addition, *A. tuberculatus*, *A. palmeri*, and *A. cruentus* had more tri-nucleotide repeats than mono-nucleotides. The presence of tri-nucleotides in the exonic region is an evolutionary mechanism to protect the genes from frame shift mutations [45]. In Amaranthaceae, a strongly biased base composition of As and Ts SSR motif was detected, such that ‘AT’, ‘AAT’, and ‘AAAT’ were more abundant with low GC-rich repeats. Similarly, di-nucleotide and tri-nucleotide repeat abundances were discovered in other plant species, including rice [46], and maize [47]. The availability of SSR primers for each species provides valuable tools for researchers in areas such as candidate gene identification, genetic diversity studies, and the establishment of phylogenetic relationships. The comparative analysis of SSRs across different species contributes to a better understanding of evolutionary relationships and genomic divergence among amaranth species.

The Identification and comparative analysis of genome-wide SNPs in five Amaranthus species provides crucial insights into genetic diversity and potential applications in crop improvement. The large-scale identification of SNPs across the entire genomes of *A. hypochondriacus*, *A. tuberculatus*, *A. palmeri*, *A. hybridus*, and *A. cruentus* signifies a comprehensive exploration of genetic variation within these species. The distinct number of SNPs identified in each species (ranging from 760,209 to 1,570,771) reflects the unique genetic makeup of individual Amaranthus species. This species-specific information is valuable for understanding the genetic diversity and potential traits that differentiate these species. The identification of specific scaffolds with higher SNP density in each species suggests potential regions of evolutionary significance or selective pressure. The percentage distribution of SNPs in genic and intergenic regions (ranging from 16.52% to 23.07% in genic regions) indicates the potential impact of these variations on coding and non-coding regions of the genome. The observation that the majority of genic SNPs are present in intronic regions emphasizes the importance of non-coding regions in contributing to genetic diversity. Hoshikawa et al. [48] performed genetic diversity analysis among Amaranthus tricolor accessions using genome-wide SNPs and found a set of 5638 SNPs without missing data in 440 accessions. Similarly, Wu and Blair [49], and Jamalluddin et al. [50] performed genetic diversity analysis and marker-trait associations in amaranths and relative species using Genotyping By Sequencing (GBS) methods. The identification of allelic variations within economically important genes highlights the practical applications of this research in crop improvement programs. These genic SNPs could serve as valuable markers for marker-assisted selection, allowing breeders to target specific traits of interest in these Amaranthus species. Comparative analysis of SNPs among the five species provides a basis for understanding evolutionary relationships and divergence. Identification of conserved or species-specific genomic regions can aid in the development of molecular tools for species differentiation and phylogenetic studies. SNPs can also be used to discover new genes and their functions by affecting gene expression and transcriptional and translational promoter activities. Therefore, they may be responsible for phenotypic variations between individuals in improving agronomic features. In plant growth and development as well as defense-related responses, Transcription Factors (TFs) operate as master key regulators. Some stress-related TF families, including WRKY, bHLH, NAC, bZIP, MADS-box, and MYB, are important TF families that control growth and developmental processes [51], biotic and abiotic stress responses [52,53], and specificity and/or crosstalk regulation between distinct TFs [54]. To regulate the expression of the related genes, TFs interact with Cis-Acting Regulatory Elements (CREs) located at the binding site or close to structural genes. Numerous CREs that are specific to different proteins involved in the initiation and regulation of transcription can be found in the promoters located upstream of a gene-encoded region [55,56]. It has been reported that the CREs exhibit a variety of functions related to biotic and abiotic components, such as light and phytohormone responsiveness, pathogen, and wound responsiveness. Studies on Cis-Regulatory Elements (CREs) are essential for a deeper understanding of how plants respond to abiotic and biotic stressors [57]. The present analysis revealed a rich diversity of TFs across the five amaranth species, with a total of 57 TF families identified based on domain sequences. The highest number of predicted TFs was observed in *A. palmeri*, suggesting potential variations in the regulatory complexity among these species. Out of the predicted 57 TF categories, bHLH, NAC, WRKY, MYB, ERF, bZIP, and M-type are the major categories similar to the earlier studies [58,59,60]. The examination of these stress-related TF families provides a foundation for understanding the genetic basis of stress responses in amaranth. The functional characterization of amaranth genes and TFs of unknown function in recent years underscores the importance of such studies for unlocking the full potential of amaranth genomes. The comparative analysis provides a foundation for future functional genomics studies, enabling researchers to prioritize candidate genes for further investigation based on their abundance and distribution patterns.

Plant miRNAs are 20–24 nucleotide long non-coding RNA sequences and have become important candidates for study due to their pivotal involvement in post-transcriptional gene regulation. The majority of miRNA targets are TFs involved in fundamental processes such as plant growth and development, response to environmental stresses, and defense mechanisms [61]. The transcripts undergo 5′ capping, splicing, and polyadenylation at the 3′ end, with subsequent processing resulting in precursor RNA (pre-miRNA) with a stem-loop structure [62]. Subsequent cleaving by DCL1 produces mature miRNA that binds to target messenger RNA (mRNA). We identified a total of 154 pre-miRNAs encoding 42 mature miRNAs across 27 miRNA families in *A. hypochondriacus*. Additionally, *A. tuberculatus* exhibited 126 pre-miRNAs encoding 36 mature miRNAs from 26 miRNA families; *A. hybridus* featured 121 pre-miRNAs with mature miRNAs from 26 miRNA families; *A. palmeri* had 110 pre-miRNAs encoding 38 mature miRNAs from 22 miRNA families; *A. cruentus* showed 119 pre-miRNAs encoding 36 mature miRNAs from 25 miRNA families using the homology search approach. Additionally, 1, 2, 0, 3, 1 one locus per each miRNA sequence were present in *A. hypochondriacus*, *A. tuberculatus*, *A. palmeri*, *A. cruentus* and *A. hybridus*, respectively. miR1320, miR2111, and miR172, for which we identified single loci in the Amaranth genome, have been identified as conserved miRNA in eudicots and play a key role in cold stress tolerance in leaves, roots, shoots, and flowers at different developmental stages [63,64]. We observed distinct loci associated with miRNAs in different Amaranthus species, namely, *A. hypochondriacus* (23), *A. tuberculatus* (20), *A. palmeri* (21), *A. cruentus* (21) and *A. hybridus* (21). This observation implies that these miRNAs emerged before Amaranth genome polyploidization, aligning with findings in other polyploid plant species [65]. miR167 and miR169 were found to be the most abundant loci across all the Amaranthus species and perform important functions in plant growth and development [66].

In plants, transporters play a crucial role in the movement of various substances across cell membranes. These transporters are integral proteins located in the cell membranes, facilitating the transport of ions, nutrients, and other essential molecules within the plant. There are different types of transporters involved in various physiological processes, such as nutrient uptake, water movement, and the maintenance of cell turgor pressure [67]. We identified transporters across five amaranth species such that transporters belonging to the protein family PF00854 (POT (proton-dependent oligopeptide transport) family) and PF00083 (sugar and other transporter) were most abundant in all the species except in *A. hypochondriacus*. Protein families PF00249 and PF00847 were observed to be abundant in *A. hypochondriacus*, comprising the Myb-like DNA-binding domain and the AP2 domain, respectively. Additionally, the transporters identified were classified based on the Transporter Classification Database (TCDB) system into well-defined classes. Moreover, 1.A.87.2.11 (Leucine-Rich Repeat (LRR) receptor-like serine/threonine-protein kinase, ERECTA) and 1.A.87.2.6 (protein BRASSINOSTEROID INSENSITIVE 1, BRI1) classes were found to be abundant in *A. hypochondriacus*, *A. tuberculatus*, and *A. hybridus*. Furthermore, 1.B.82.1.3 class (uncharacterized protein) and subsequently 1.A.87.2.11 were observed to be abundant in *A. palmeri* and *A. cruentus*.

## 4. Materials and Methods

### 4.1. Data Sources and Sequence Retrieval

The genome sequence data of *A. hypochondriacus* were retrieved from PhytozomeV13 (https://phytozome.jgi.doe.gov/Ahypochondriacus_er; accessed on 4 May 2022), *A. tuberculatus*, *A. hybridus*, *A. palmeri* from a Comparative Genomics Platform CoGe: (https://genomevolution.org/coge/; Genome ID 54057, 57429, 56750; accessed on 5 May 2022), and of *A. cruentus* from the NCBI database repository (https://www.ncbi.nlm.nih.gov/genome/109717?genome_assembly_id=1765032; accessed on 5 May 2022). The genome assembly of *A. hypochondriacus*, *A. tuberculatus*, *A. hybridus*, and *A. palmeri* was assembled scaffold wise, and *A. cruentus* was assembled into 17 pseudomolecules (Table 8).

### 4.2. Assembly Quality Assessment and Annotation

The completeness of the selected amaranth genomes was assessed using the Benchmarking Universal Single-Copy Orthologs (BUSCO) v.4.0 [68], which contained 2326 genes. For evaluation and completeness of genes in the assemblies, unigenes were generated from the available transcriptomic data of different amaranth species using Trinity v.2.15.1 software with the parameters, kmer length of 25, and min kmer coverage of 2 [69] and then mapped with the genome assemblies using CLC Genomics Workbench v.24.0 [70]. For repetitive sequence analysis, we used homology-based approach. RepeatModeler v.2.0.3 and RepeatMasker v.4.1.3 (http://www.repeatmasker.org; accessed on 17 October 2023) were used to identify and characterize repeats based on the RepBase v.27.06 and Dfam reference libraries (http://www.girinst.org/repbase; accessed on 18 October 2023) and a custom library obtained through careful self-training. In this analysis, RepeatScout v.1.0.5, a module of RepeatModeler software was used to identify a set of repeat elements that was further utilized by Recon v.1.08, another module of the same software to generate a classified consensus repeat library. The merged repeat library was subjected to RepeatMasker v.4.1.3 for homology-based masking of the repeat regions in targeted amaranth genome sequences.

### 4.3. Gene Structure Analysis and Functional Annotation

Gene structure features such as mean exons per gene, mean exon length (bp), mean introns per gene, and mean intron length (bp) were analyzed using in-house scripts. Functional annotation of protein-coding genes was based on sequence similarity and domain conservation was performed using BLASTP searches of protein-coding genes against different public databases such as NCBI-NR, Swiss-Prot, KEGG, InterProScan, GO, TrEMBL, and COG databases with E-value cutoff < 1 × 10^−5^. InterProScan analysis gives putative gene function with different databases, including pFam, Gene3D, PANTHER, CDD, SUPERFAMILY, PRINTS, SMART, and ProSite.

### 4.4. Gene Family Analysis and Phylogenetic Distribution

Gene family relationships were assigned using OrthoMCL v.2.0.9 software [71] between five amaranth species and five other species: *A. thaliana* (https://ftp.ensemblgenomes.ebi.ac.uk/; accessed on 8 November 2023), *B. vulgaris* (https://phytozome-next.jgi.doe.gov/info/Bvulgarisssp_vulgaris_EL10_2_2; accessed on 8 November 2023), *C. quinoa* (https://phytozome-next.jgi.doe.gov/info/Cquinoa_v1_0; accessed on 8 November 2023), *S. oleracea* (http://www.spinachbase.org/; accessed on 8 November 2023), and *S. aralocaspica* (http://aspera.gigadb.org/dataset/view/id/100646/; accessed on 8 November 2023) based on an all-vs.-all BLASTP alignment with an E-value cutoff of 1 × 10^−5^. We constructed a phylogenetic tree based on single-copy orthologs using the maximum likelihood approach in rAxML v.8.2.4 [72]. The single-copy ortholog sequences were aligned with MUSCLE v.5.1 [73]. To trim the alignment to make all the aligned sequences of the same length, we used trimAl tool v.1.2 [74]. Now, we have trimmed the alignment in PHYLIP format, which will be used to concatenate and generate a supermatrix using FASconCAT perl script [75]. Furthermore, ProtTest v.3.4.2 [76] was used to estimate the best-fitting substitution model. The estimation of the best evolutionary model is determined by various frameworks, such as Akaike Information Criterion (AIC), Bayesian Information Criterion (BIC), Second-Order Akaike framework (AICc), and Decision Theory (DT). The best predicted evolutionary model was JTT + I + G + F. With this alignment and model, we reconstructed the phylogeny using RA × ML and 100 bootstrap repetitions.

### 4.5. Simple Sequence Repeats (SSRs) Identification

Microsatellite identification from *A. hypochondriacus*, *A. tuberculatus*, *A. hybridus*, *A. palmeri*, and *A. cruentus* genomes was performed using Krait tool v.1.3.3 (https://github.com/lmdu/krait; accessed on 12 June 2022) [77] with the parameters such as twelve repeat units for mono-nucleotide, six repeat units for the di-nucleotide, and five repeat units for tri-nucleotide to hexa-nucleotides [78,79]. The maximum difference between the two SSRs is 100 bp. Primer3 v.3.2.0 was used to design primers from the flanking region of each identified SSR with the following parameters: primer length, 18–27 bp; PCR product size, 150–300 bp, with an optimum of 180 bp; melting temperature, 55–65 °C; GC content, 40–60%, with an optimal value of 50% [80]. In-house Perl scripts were used to identify gene-specific SSRs and the distribution of different types of SSRs in the genic and intergenic regions.

### 4.6. SNPs and TFs Identification

RNA-Seq datasets were obtained from the NCBI SRA database with bioproject numbers PRJNA290898, PRJNA432348, and PRJNA626536 for *A. hypochondriacus*, *A. tuberculatus*, and *A. palmeri*, respectively. Mining of high-quality SNPs involves several steps using various bioinformatic tools. The preliminary step is to perform a quality control check and trimming of adapter sequences from raw sequencing reads data using Trimmomatic v.0.39, a flexible trimmer for Illumina paired-end data [81]. The next step is the mapping of processed high-quality reads with corresponding amaranth genomes using the Burrows-Wheeler Aligner (BWA) with default parameters [82]. The read-mapped alignment file is in SAM format. SAMTools v.1.19.1 was used to convert SAM to BAM file format, and then for shortening of BAM file by removing the duplicate reads [83]. SNP calling and filtering were performed using SAMTools VarScan, and bcftools v.1.19. We used an in-house Perl script to extract the final high-quality SNPs and their location within the gene region, as well as 3′ and 5′ flanking sequences. Transcription factor families in *A. hypochondriacus*, *A. tuberculatus*, *A. hybridus*, *A. palmeri*, and *A. cruentus* were identified by BLASTX similarity search of the protein-coding genes of the respective Amaranth species against plant transcription factor database (PlantTFDB v.4.0) [84] with the parameters bit score > 100 and e-value 1 × 10^−5^ [85]. The functional annotation of the predicted TFs was performed using InterProScan 5 [86].

### 4.7. miRNA and Transporter Gene Identification

Already known and mature miRNAs were obtained from PMRD-plant microRNA database (http://bioinformatics.cau.edu.cn/PMRD; accessed on 22 May 2023) and aligned with the *A. hypochondriacus*, *A. tuberculatus*, *A. hybridus*, *A. palmeri*, and *A*. *cruentus* genomes using BLASTN search with no mismatches. The database consists of a total of 2618 mature miRNA sequences, including *Arabidopsis thaliana*, *Glycine max*, *Oryza sativa*, *Solanum lycopersicum*, *Triticum aestivum*, and *Zea mays*. We extracted 200 nucleotides upstream and downstream sequences surrounding every matched region and discarded the miRNA candidates coded for proteins/lies within the CDS region or repetitive elements to produce a possible miRNA set. The UNAFold web server (http://www.unafold.org/; accessed on 24 June 2022) was used to generate and evaluate the potential secondary structures of pre-miRNAs. The Minimum Free Energy (MFE) values were calculated for each secondary structure, and filtered secondary structure with the lowest MFE values [87]. Target identification of the predicted candidate miRNAs was carried out using the online tool psRNAtarget (http://plantgrn.noble.org/psRNATarget/; accessed on 24 June 2022). The predicted miRNA target genes were annotated using the online agriGO tool (http://bioinfo.cau.edu.cn/agriGO/; accessed on 26 June 2022). Transporter genes in *A. hypochondriacus*, *A. tuberculatus*, *A. hybridus*, *A. palmeri*, and *A. cruentus*, were identified using BLASTP search of the protein sequences of respective amaranth species against Transporter Classification Database (TCDB; http://www.tcdb.org/; accessed on 26 June 2022), with the following parameters: E-value and Bit Score cutoffs of 10^−5^ and 100, respectively. The Transmembrane (TM) domains were analyzed using TMHMM Server v.2.0 [88]. The transporter genes with (TM) domains greater than two were extracted and classified in different sub-families using the TCDB database.

## 5. Conclusions

In this study, we performed repeat analysis of these amaranth genomes which provides valuable insights into the complexity and diversity of their repetitive sequences. The analysis of gene distribution and functional annotation in these amaranth species provides a rich resource for researchers, opening avenues for further exploration of the molecular mechanisms underlying their diverse traits and adaptation strategies. The detailed investigation of gene clustering and phylogenetic analysis in this study provides a comprehensive view of the genomic landscape of amaranth species and their evolutionary relationships and has practical implications for crop improvement and biodiversity conservation within the Amaranthaceae family. We also performed a comparative analysis of SSRs, SNPs, and TFs, in five amaranth species providing a foundation for a deeper understanding of their genetic makeup. The insights gained from this study have practical applications in genetic research and agriculture, and potentially contribute to the development of improved varieties through selective breeding and molecular marker-assisted breeding programs. And this provides a valuable resource for future studies aimed at deciphering the molecular mechanisms underlying stress tolerance, adaptation, and the diverse biological processes in these agriculturally important plants. These findings contribute not only to basic genomic knowledge, but also hold potential applications in areas like crop genetics, genomics, evolution, and breeding of amaranth as well as many other closely-related amaranth relatives.

## Figures and Tables

**Figure 1 plants-13-00824-f001:**
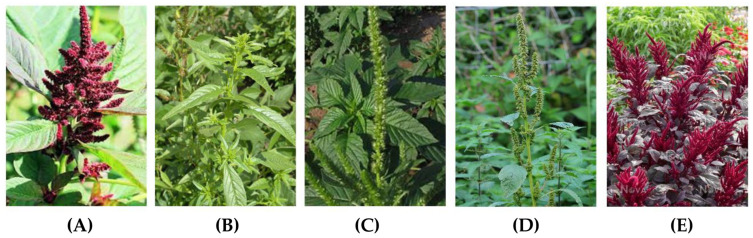
Inflorescences of selected amaranth species. (**A**) Dense inflorescence of *A. hypochondriacus*, (**B**) inflorescence of *A. tuberculatus*, (**C**) terminal inflorescence of *A. palmeri*, (**D**) elongated inflorescence of *A. hybridus*, and (**E**) terminal inflorescence of *A. cruentus*. Images source: amaranth genomic resource database.

**Figure 2 plants-13-00824-f002:**
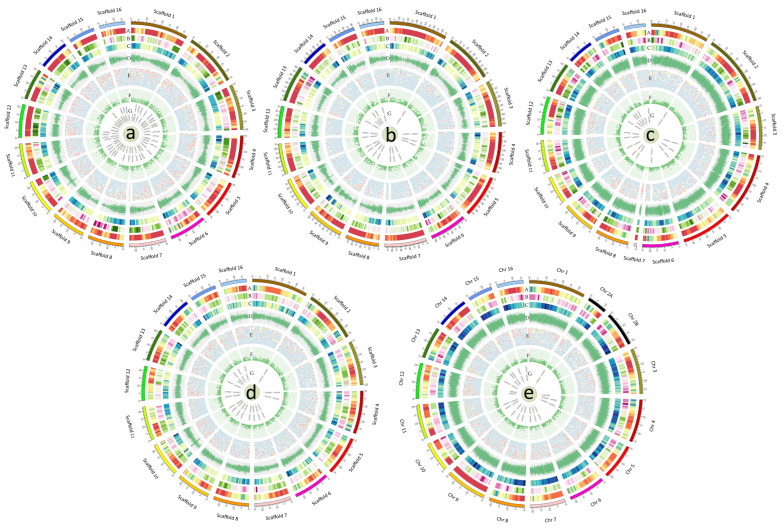
Circos plots showing genomic features in (**a**) *A. hypochondracus*, (**b**) *A. tuberculatus*, (**c**) *A. palmeri*, (**d**) *A. hybridus*, and (**e**) *A. cruentus*. The genomic feature in the concentric circles indicates the sixteen scaffolds in (Figure 1A–D), and seventeen chromosomes in (Figure 1E). The tracks from outer to inner circles indicate A: the density of genic SNPs in 1-Mb windows; B: density of intergenic SNPs in 1-Mb windows; C: density of TFs in 1-Mb windows; D: distribution of protein-coding genes; E: distribution of (mono- to hexa-) type of SSRs; F: transporter genes; G: putative miRNAs.

**Figure 3 plants-13-00824-f003:**
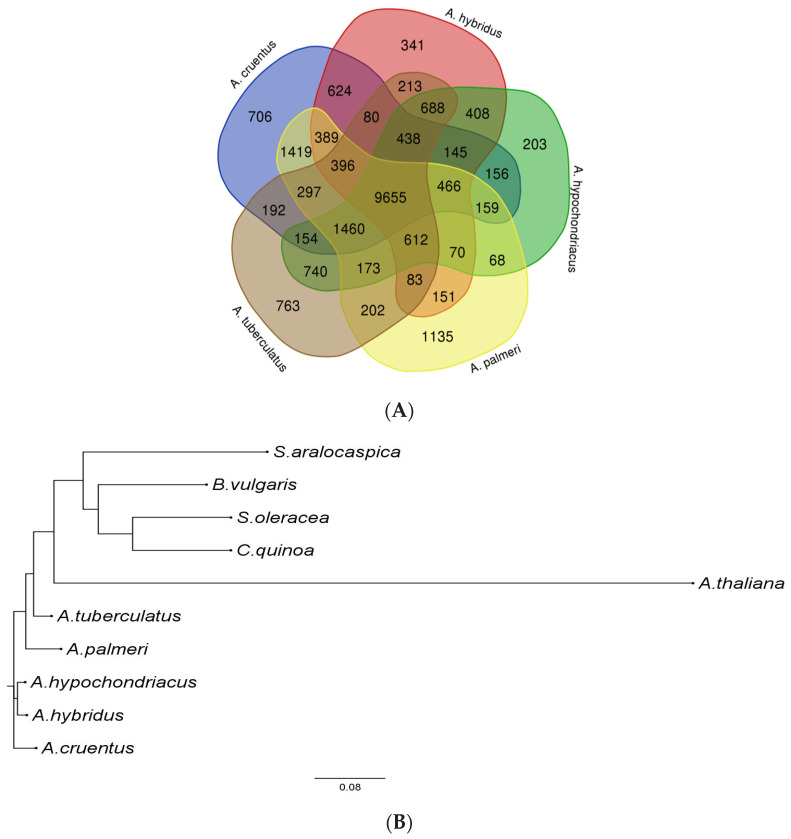
The groups of orthologs shared by the Amaranthaceae species. (**A**) Groups of orthologs shared between *A. hypochondriacus*, *A. tuberculatus*, *A. palmeri*, *A. hybridus*, and *A. cruentus*. Venn diagram was generated from Bioinformatics & Evolutionary Genomics portal. (**B**) Phylogeny of the concatenated dataset using 406 single-copy orthologs extracted from five amaranth species, and five other genomes. The *A. thaliana* was taken as an out-group in this analysis.

**Figure 4 plants-13-00824-f004:**
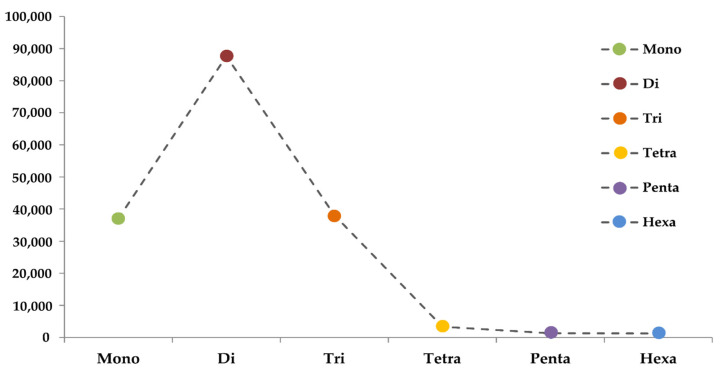
The average number of six SSR types (mono- to hexa-nucleotides) in five amaranth genomes.

**Figure 5 plants-13-00824-f005:**
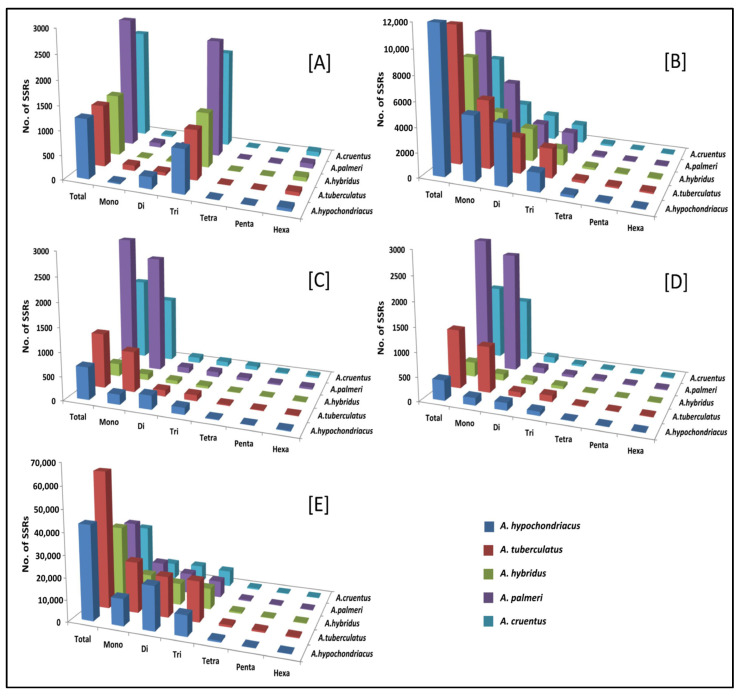
Distribution of SSRs (mono- to hexa-nucleotide) in different genic (exon, intron, 5′UTRs, 3′UTRs) and intergenic regions of five amaranth species. (**A**–**E**) represents exons, introns, 5′UTRs, 3′UTRs, and intergenic regions, respectively.

**Figure 6 plants-13-00824-f006:**
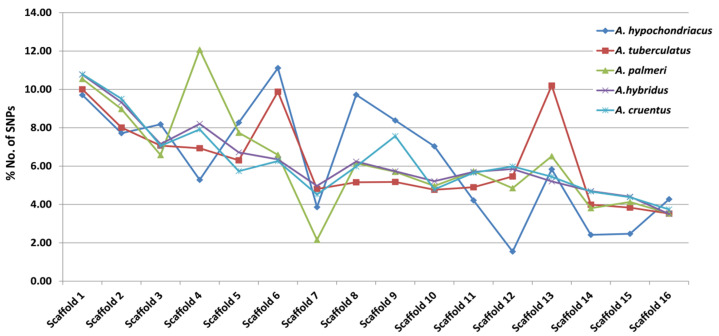
Scaffold-wise distribution of SNPs across *A. hypochondriacus*, *A. tuberculatus*, *A. palmeri*, *A. hybridus*, and *A. cruentus* genomes.

**Figure 7 plants-13-00824-f007:**
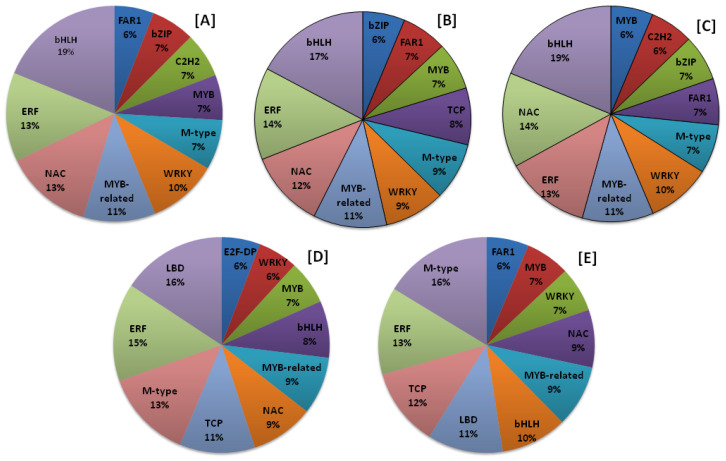
Distribution of top ten TF categories across five amaranth plant species. (**A**) *A. hypochondriacus*, (**B**) *A. tuberculatus*, (**C**) *A. palmeri*, (**D**) *A. hybridus*, and (**E**) *A. cruentus*.

**Figure 8 plants-13-00824-f008:**
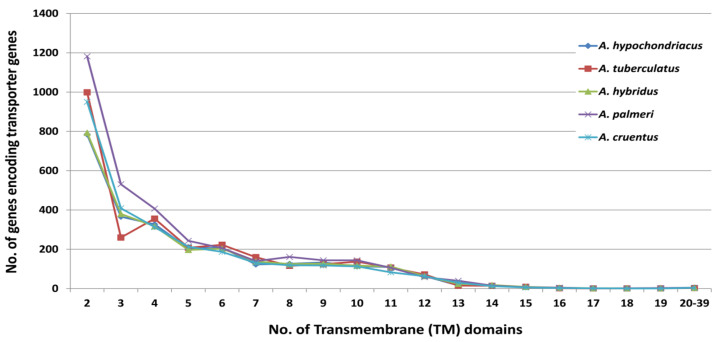
Distribution of TM domains across the peptide sequences of five amaranth species.

**Table 1 plants-13-00824-t001:** Completeness evaluation of genome assembly using BUSCO database in five amaranth species.

BUSCO	*A. hypochondiacus*		*A. tuberculatus*		*A. palmeri*		*A. hybridus*		*A. cruentus*	
	No.	%	No.	%	No.	%	No.	%	No.	%
**Complete single copy**	2100	90.3%	2068	88.9%	1921	82.6%	2131	91.6%	2063	88.7%
**Complete duplicated**	98	4.2%	160	6.9%	286	12.3%	102	4.4%	91	3.9%
**Fragmented**	26	1.1%	12	0.5%	14	0.6%	14	0.6%	14	0.6%
**Missing**	102	4.4%	86	3.7%	105	4.5%	79	3.4%	158	6.8%
**Total**	2326	100%	2326	100%	2326	100%	2326	100%	2326	100%

**Table 2 plants-13-00824-t002:** Distribution of different classes of repeats (%) in *A. hypochondriacus*, *A. tuberculatus*, *A. palmeri*, *A. hybridus*, and *A. cruentus* genome sequences.

Species	*A. hypochondriacus*			*A. tuberculatus*			*A. palmeri*			*A. hybridus*			*A. cruentus*		
Repeat Class	No. of Repeats	Length (bp)	% in Genome	No. of Repeats	Length (bp)	% in Genome	No. of Repeats	Length (bp)	% in Genome	No. of Repeats	Length (bp)	% in Genome	No. of Repeats	Length (bp)	% in Genome
**SINEs**	13,578	1,934,332	0.48%	12,373	1,716,321	0.25%	6042	844,099	0.20%	9515	1,361,744	0.33%	10,516	1,630,367	0.45%
**LINEs**	36,586	15,775,607	3.91%	71,876	41,301,986	5.99%	45,767	20,834,341	5.06%	50,434	15,762,835	3.83%	30,237	11,998,143	3.29%
**LTR Elements**	158,661	78,481,863	19.43%	155,302	171,464,524	24.89%	149,558	73,936,501	17.95%	94,237	57,299,089	13.91%	154,295	76,468,935	20.94%
**DNA Transposons**	118,098	33,996,053	8.42%	166,488	55,149,055	8.00%	112,877	27,436,958	6.66%	126,739	37,471,073	9.10%	104,357	30,847,128	8.45%
**Small RNA**	4064	640,236	0.16%	4846	1,366,752	0.20%	4753	900,858	0.22%	4424	1,252,736	0.30%	1870	382,684	0.10%
**Sattelites**	2339	302,661	0.07%	2034	677,293	0.10%	116	32,271	0.01%	0	00	0.0%	70	34,509	0.01%
**Simple repeats**	107,707	11,161,024	2.76%	151,290	12,217,687	1.77%	106,058	16,481,473	4.00%	99,630	8,901,154	2.16%	88,116	5,915,869	1.62%
**Low complexity**	17,859	930,846	0.23%	23,430	1,257,471	0.18%	19,052	1,019,810	0.25%	16,497	838,740	0.20%	15,962	811,218	0.22%
**Others**	434,419	85,037,944	21.05%	638,459	148,260,761	21.52%	432,867	82,052,081	19.92%	488,378	109,052,159	26.48%	398,308	77,599,307	21.25%
**Total**	893,311	229,717,046	56.88%	1,226,098	435,846,425	63.26%	877,090	224,470,421	54.49%	889,854	233,584,532	56.72%	803,731	207,438,445	56.80%

**Table 3 plants-13-00824-t003:** Gene structure features of *Amaranthus hypochondriacus*, *Amaranthus tuberculatus*, *Amaranthus palmeri*, *Amaranthus hybridus*, *Amaranthus cruentus*, *Suaeda aralocaspica*, *Chenopodium quinoa*, *Spinacea oleracea*, and *Beta vulgais*.

Species	No. of Protein-Coding Genes	Mean Gene Length (bp)	Mean CDS Length (bp)	Mean Exons per Gene	Mean Exon Length (bp)	Mean Introns per Gene	Mean Intron Length (bp)
*A. hypochondriacus*	23,879	4102	1066	5	219	4	786
*A. tuberculatus*	30,771	4304	941	4	303	3	787
*A. palmeri*	48,625	4371	1486	4	385	3	394
*A. hybridus*	23,820	4472	1170	4	267	3	819
*A. cruentus*	43,382	4154	1356	5	346	4	386
*S. aralocaspica*	29,604	4463	1117	5	234	4	891
*C. quinoa*	44,776	4797	1274	6	264	5	671
*S. oleracea*	25,495	5716	1156	5	277	4	908
*B. vulgaris*	27,421	4302	1057	4	236	3	937

**Table 4 plants-13-00824-t004:** Statistical analysis of the functional annotations of protein-coding genes in the genomes of *A. hypochondriacus*, *A. tuberculatus*, *A. palmeri*, *A. hybridus*, and *A. cruentus*.

Name of Database	*A. hypochondriacus*		*A. tuberculatus*		*A. palmeri*		*A. hybridus*		*A. cruentus*	
	No. of Genes	Percentage (%)	No. of Genes	Percentage (%)	No. of Genes	Percentage (%)	No. of Genes	Percentage (%)	No. of Genes	Percentage (%)
**NCBI-NR**	22,270	93.26%	24,554	79.79%	41,582	85.51%	20,333	85.36%	37,393	86.19%
**Swiss-Prot**	16,887	70.71%	18,822	61.16%	22,416	46.09%	14,793	62.10%	24,805	57.17%
**KEGG**	6269	26.25%	3148	10.23%	7559	15.54%	5661	23.76%	4172	9.61%
**InterPro-Scan**	21,517	90.10%	23,028	74.83%	38,762	79.71%	19,677	82.60%	34,475	79.46%
**GO**	6118	25.62%	5937	19.29%	6231	12.81%	5847	24.54%	6240	14.38%
**TrEMBL**	21,800	91.29%	23,496	76.35%	39,400	81.02%	19,778	83.03%	35,337	81.45%
**COG**	9657	40.44%	10,933	35.53%	13,209	27.17%	8936	37.51%	11,181	25.77%
**Total**	22,753	95.28%	27,309	88.75%	42,889	88.20%	20,973	88.05%	38,654	89.10%
**Un-annotated**	1126	4.72%	3462	11.25%	5736	11.80%	2847	11.95%	4728	10.90%

**Table 5 plants-13-00824-t005:** A comprehensive survey of SSR motifs (mono- to hexa-types) identification across the five amaranth genomes.

Species	Genome Size (Mb)	No. of SSRs	Relative Abundance (loci/Mb)	Relative Density (bp/Mb)
*A. hypochondriacus*	403.89	243,288	602.43	16,894.21
*A. tuberculatus*	688.98	216,733	331.6	12,151.11
*A. hybridus*	411.83	132,717	329.35	12,992.27
*A. palmeri*	411.92	144,801	355.22	14,857.37
*A. cruentus*	365.2	101,040	276.77	8707.26
**Total**	**2281.82**	**838,579**	**367.5**	**12,815.54**

**Table 6 plants-13-00824-t006:** Distribution of three predicted sets of SSR primer pair (mono- to hexa-nucleotide) types in five amaranth genomes.

Species	Total SSRs	Mono	Di	Tri	Tetra	Penta	Hexa
*A. hypochondriacus*	57,404	17,907	25,789	12,117	967	286	338
*A. tuberculatus*	77,814	30,814	21,836	22,365	1249	868	682
*A. hybridus*	50,059	23,246	12,857	12,293	916	315	432
*A. palmeri*	41,930	16,303	10,422	13,400	862	417	526
*A. cruentus*	36,462	13,075	10,682	11,301	848	231	325
**Total**	**263,669**	**101,345**	**81,586**	**71,476**	**4842**	**2117**	**2303**

**Table 7 plants-13-00824-t007:** Putative miRNAs and their target genes were identified in five amaranth species by genome-wide analysis.

Species	No. of Pre-miRNAs	Mature miRNAs	miRNA Families	No. of Targets
*A. hypochondriacus*	123	31	22	512
*A. tuberculatus*	94	27	22	869
*A. hybridus*	113	32	22	707
*A. palmeri*	109	31	21	659
*A. cruentus*	80	32	24	365

**Table 8 plants-13-00824-t008:** Summary of the genome sequence data of different amaranth species used in this study.

Species	Assembly Version	Assembly Level	Genome Size (Mb)	GC %
*A. hypochondriacus*	Ver.2.0 (Phytozome)	Scaffold	403.89	32.65%
*A. tuberculatus*	Ver.2.0 (CoGe)	Scaffold	688.98	33.14%
*A. hybridus*	Ver.1.0 (CoGe)	Scaffold	411.83	32.30%
*A. palmeri*	Ver.1.2 (CoGe)	Scaffold	411.92	33.18%
*A. cruentus*	Ver.1.0 (NCBI)	Chromosome	365.20	33.03%

## Data Availability

All data supporting the findings of this study are available with the manuscript and its Appendix A.

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
