# Peer review of "Genome-Wide Comparative Analysis of Five Amaranthaceae Species Reveals a Large Amount of Repeat Content"

_plants, 2024, doi:10.3390/plants13060824_

Round 1

Reviewer 1 Report

Comments and Suggestions for Authors

This article provides a detailed analysis of the genomes of five amaranth species, but I think there are still some changes that need to be made.

1. There are too many descriptions of amaranth functions in the introduction section.

2. Keywords and Abstract are inappropriate and do not help people understand the content and innovation of the article.

3. Figure 1 is not clear.

4. What are the characteristics of these five amaranthus varieties and their corresponding pictures should be described.

5. Figure 2B is meaningless.

Comments on the Quality of English Language

The English level needs to be further polished.

Author Response

This article provides a detailed analysis of the genomes of five amaranth species, but I think there are still some changes that need to be made.

Reply: Thanks for your appreciation. Your valuable suggestions are very important to improve the quality of our manuscript.

Comment 1. There are too many descriptions of amaranth functions in the introduction section.

Reply: The descriptions of amaranth functions in the introduction section have been shortened in the revised version of the manuscript, as suggested.

Comment 2. Keywords and Abstract are inappropriate and do not help people understand the content and innovation of the article.

Reply: Some potential keywords have been added to the keywords section of the revised manuscript at lines 34-35.

Comment 3. Figure 1 is not clear.

Reply: Figure 1 has been revised and incorporated into the revised version of the manuscript.

Comment 4. What are the characteristics of these five amaranthus varieties and their corresponding pictures should be described.

Reply: The characteristics of studied amaranth species with their corresponding images have been incorporated in the revised manuscript, from lines 115-141.

Comment 5. Figure 2B is meaningless.

Reply: Figure 2B has been moved to supplementary materials in the revised manuscript.

Comment 6. The English level needs to be further polished.

Reply: The manuscript has been thoroughly checked for English and grammatical errors and corrected in the revised version of the manuscript as suggested by the reviewer.

Reviewer 2 Report

Comments and Suggestions for Authors

This is a very interesting and well-written manuscript. 

The presentation of data and discussion is nicely documented, while the figures and tables are a great addition to the results section.

I would only suggest a shorter title that would also indicate the most important result of the study.

"Genome-wide comparative analysis of five Amaranthaceae species reveals a large amount of repeat content.

Author Response

This is a very interesting and well-written manuscript. The presentation of data and discussion is nicely documented, while the figures and tables are a great addition to the results section.

Reply: Thanks for your appreciation.

Comment 1. I would only suggest a shorter title that would also indicate the most important result of the study. "Genome-wide comparative analysis of five Amaranthaceae species reveals a large amount of repeat content.

Reply: The title of the manuscript has been shortened to “Genome-wide comparative analysis of five Amaranthaceae species reveals a large amount of repeat content”, as suggested.

Reviewer 3 Report

Comments and Suggestions for Authors

Dear authors,

the manuscript presents an exstensive bioinformatic work aimed at elucidating the genomic structure and gene content of five amaranth species, namely Amaranthus hypochondriacus, Amaranthus tuberculatus, Amaranthus hybridus, Amaranthus palmeri, and Amaranthus cruentus, through whole-genome comparative analysis.

The  findings are likely to be of interest to researchers in the fields of genetics, genomics, evolution, and agriculture. Microsatellite mining and primer design and SNP discovery were also performed, providing molecular tools which may be useful for further genetic studies and breeding purposes.

Below are some suggestions to improve the presentation and clarity of your work.

The introduction is comprehensive and up-to-date, incorporating relevant background information on the genus Amaranthus, its historical significance, utilization, and potential applications.

I only suggest to avoid redundancy and repetition of some concepts that are confounding, such as lines 48 49 that are a useless repetition of lines 42 44.

Please check also line 66 (closing bracket is missing)

The results presented appear to be comprehensive and methodologically sound. 

I have only a few suggestions:

In section 2.3 Gene content, distribution and functional annotation

The authors should comment and discuss the unnotated portion of the genes of Table 4; 

Also, the results from line 199 to 204, should be presented in a more clear way, easier to follow.

In section 2.4 Gene family construction and phylogenetic distribution

The presentation of the results in lines 214-222, should be improved, because it appears unclear for a reader which is not familiar with the analysis and some data seem to be contrasting.

Comments on the Quality of English Language

The language is generally clear and precise, however, some sentences should be rephrased  to enhance clarity and flow, as above suggested to the authors.

Author Response

The manuscript presents an exstensive bioinformatic work aimed at elucidating the genomic structure and gene content of five amaranth species, namely Amaranthus hypochondriacusAmaranthus tuberculatusAmaranthus hybridusAmaranthus palmeri, and Amaranthus cruentus, through whole-genome comparative analysis.

The findings are likely to be of interest to researchers in the fields of genetics, genomics, evolution, and agriculture. Microsatellite mining and primer design and SNP discovery were also performed, providing molecular tools which may be useful for further genetic studies and breeding purposes.

Below are some suggestions to improve the presentation and clarity of your work.

Reply: Thanks for your appreciation. Your valuable suggestions are very important to improve the quality of our manuscript.

Comment 1. The introduction is comprehensive and up-to-date, incorporating relevant background information on the genus Amaranthus, its historical significance, utilization, and potential applications.

Reply: Thank you for your appreciation and valuable suggestions.

Comment 2. I only suggest to avoid redundancy and repetition of some concepts that are confounding, such as lines 48 49 that are a useless repetition of lines 42 44.

Reply: The repetitive lines 50-52 have been removed in the revised version of the manuscript as suggested.

Comment 3. Please check also line 66 (closing bracket is missing)

Reply: At line 69 of the revised manuscript, the missing closing bracket has been inserted.

The results presented appear to be comprehensive and methodologically sound. 

I have only a few suggestions:

Comment 4. In section 2.3 Gene content, distribution and functional annotation

The authors should comment and discuss the unnotated portion of the genes of Table 4; 

Reply: Discussion about unannotated genes of Table 4 has been incorporated from lines  234-236 in the revised manuscript, as suggested by the reviewer.

Comment 5. Also, the results from line 199 to 204, should be presented in a more clear way, easier to follow.

Reply: Corrections have been incorporated from lines 236-242 in the revised manuscript, as suggested by the reviewer.

Comment 6. In section 2.4 Gene family construction and phylogenetic distribution

The presentation of the results in lines 214-222, should be improved, because it appears unclear for a reader which is not familiar with the analysis and some data seem to be contrasting.

Reply: In section 2.4, the results have been improved from lines 253-262 in the revised version of the manuscript, as suggested by the reviewer.

Comment 7. The language is generally clear and precise; however, some sentences should be rephrased to enhance clarity and flow, as above suggested to the authors.

Reply: The manuscript has been thoroughly checked for English and grammatical errors and corrected in the revised version of the manuscript as suggested by the reviewer.